# Effects of Linseed Meal and Carotenoids from Different Sources on Egg Characteristics, Yolk Fatty Acid and Carotenoid Profile and Lipid Peroxidation

**DOI:** 10.3390/foods10061246

**Published:** 2021-05-31

**Authors:** Tatiana D. Panaite, Violeta Nour, Mihaela Saracila, Raluca P. Turcu, Arabela E. Untea, Petru Al. Vlaicu

**Affiliations:** 1Laboratory of Chemistry and Nutrition Physiology, National Research Development Institute for Animal Biology and Nutrition, Calea Bucuresti nr. 1, Balotesti, 077015 Ilfov, Romania; mihaela.saracila@ibna.ro (M.S.); raluca.socoliuc@ibna.ro (R.P.T.); arabela.untea@ibna.ro (A.E.U.); alexandru.vlaicu@outlook.com (P.A.V.); 2Department of Horticulture & Food Science, University of Craiova, 13 AI Cuza Street, 200585 Craiova, Romania

**Keywords:** kapia pepper, sea buckthorn pomace, carrot, yolk color, fatty acids, carotenoids, cholesterol, lipid oxidation

## Abstract

The present study aimed to evaluate the effect of supplementing the diet of laying hens with linseed meal and carotenoids from different sources on egg characteristics, yolk fatty acid and carotenoid profile, and lipid peroxidation. A 4-week experiment was conducted on 168 Lohmann Brown layers (43 weeks of age), assigned to four dietary treatments (42 hens/group; 21 replicate/groups with 2 birds/pen) consisting of a control diet (C) and three diets simultaneously supplemented with 6% linseed meal and 2% dried kapia pepper (E1), 2% dried sea buckthorn pomace (E2) and 2% dried carrot (E3). Every 2 weeks, 18 eggs/group/period were collected randomly from each group and used to determine the egg quality and nutritional parameters. The results showed that dietary linseed meal and carotenoids sources improved egg color, carotenoids’ accumulation in egg yolk and fatty acid profile, especially the n-3 PUFA content. Dietary carotenoids supplementation reduced, n-6/n-3 ratio, cholesterol content of the egg yolk and improved yolk pH, egg thickness and yolk oxidative stability. In conclusion, the use of these sources of carotenoids in the linseed meal enriched diets could be an effective way to improve the nutritional properties of the eggs without affecting their quality and consumer’s safety.

## 1. Introduction

Nowadays, the sensory and nutritional quality of eggs is becoming a growing concern for consumers. Also, apart from the price, the physical characteristics such as the size of the eggs, the color of the yolk but also the freshness represent important criteria for the consumer when purchasing eggs. Currently, there is a growing demand for eggs enriched in nutrients such as polyunsaturated fatty acids, vitamins (D, E, etc.), minerals (selenium, iron, zinc, etc.), antioxidant compounds, as this enrichment can improve the health status and well-being of consumers [1]. Moreover, according to many researchers, PUFA/SFA, PUFA n-6/n-3 ratios and atherogenic and thrombogenic indices are currently polyunsaturated fatty acids/saturated fatty acids (PUFA/SFA) among the most important indices for assessing the nutritional value of foods and the impact on consumer’s health [2,3].

In the last decade, among the most used compounds for enriching eggs are omega-3 fatty acids and carotenoids [4,5]. Therefore, to ensure the above, the nutrition of laying hens plays a key role. It is well-known that the fatty acid composition of the hen’s diet influences the fatty acid composition of eggs [6,7]. The flaxseed has been intensively used as a natural source of omega 3 in hen’s diet. Of the total fatty acids in flaxseed, 53% are α-linolenic acid (ALA), 17% linoleic acid (LA), 19% oleic acid, 3% stearic acid and 5% palmitic acid [8]. The disadvantage of egg enrichment in omega-3 polyunsaturated fatty acids is the increased susceptibility of yolk lipid peroxidation, through that is affected both the nutritional and sensory quality of eggs and the safety of consumers. Therefore, simultaneously enrichment of eggs with antioxidant compounds (vitamin E and carotenoids) was reported to reduce fatty acid oxidation and provide a good source of dietary antioxidants [9,10]. The dietary antioxidants are preferentially stored by the laying hen in the egg [11]. In addition, carotenoids are red and yellow pigments which influence egg yolk colour [12]. Moreover, data showed that carotenoids intake is associated with a lower incidence of cardiovascular disease due to their mechanisms such as free radicals’ scavenger and low-density lipoprotein cholesterol resistance to oxidation inductors [13,14]. Thus, the addition of carotenoids sources in hen’s diet has two potential actions: as natural pigment to improve egg yolk color and quality, and as antioxidant to delay the oxidation of polyunsaturated fatty acids. The use of natural carotenoids is consistent with the preference of many consumers for natural and safety products [12].

Red pepper, carrot and sea buckthorn are important sources of natural antioxidants and carotenoids, that can prevent oxidative reactions of PUFA [1,15,16]. Red kapia pepper (*Capsicum annuum* L.) is widespread throughout the world as vegetable [17,18]. It is a rich source of carotenoids [12,19], ranging from 0.3 to 3.2 g/100 g dry weight [20], vitamins C and E [21,22]. Many researchers have found that dietary red pepper improved the intensity of egg yolk color [23,24], the hen’s feed intake [17], and the rate of laying [23,25].

Carrot (*Daucus carota*) is a root vegetable rich in minerals, fiber, carbohydrates, antioxidant flavonoids, most of essential micronutrients, and especially beta-carotene [26]. It contains higher amount of carotenes and less amount of xanthophylls [15]. Due to its low economic value, low quality carrot and its discarded parts can be used as carotenoid source in animal feed. Spasevski et al. [15] reported that dietary dried carrot had no negative effects on physical characteristics of eggs, egg yolk color, egg acceptability or β-carotene content.

Sea buckthorn (*Hippophae rhamnoides* L.) is a rich source of nutrients and bioactive compounds such as vitamins, amino acids, phytosterols, polyunsaturated fatty acids, carotenoids and phenolics compounds [27,28,29]. However, a little attention was given to their by-products. Nour et al. [30] showed that sea buckthorn pomace contains high level of essential nutrients such as protein, crude fat, crude fiber, amino acids, (arginine, leucine, valine, threonine, phenylalanine), alpha-linolenic acid, minerals and phenolic compounds, the last ones leading to a high antioxidant capacity. Thus, the valuable composition and the lower price compared to the fruits makes it worth studying as an ingredient in the diet of laying hens. Many authors suggested that dietary sea buckthorn pomace increased the color of the egg yolk, and significantly affected the total number of laid eggs [31,32]. 

To our knowledge, no studies have been performed to investigate the implications of supplementing diets enriched with n-3 PUFA with these carotenoid sources. In this context, the aim of this study was to investigate the effect of using red kapia pepper, carrot and sea buckthorn pomace to the linseed meal supplemented feed for laying hens on the performance, physical and internal egg quality, and oxidative stability of the egg yolk.

## 2. Materials and Methods

### 2.1. Materials

The by-products from sea buckthorn processing were obtained from Biocat Prod S.R.L., a buckthorn producer and processor from Grădina, Constanța county, Romania. Kapia peppers and carrots were purchased from the local market. They were dehydrated in an industrial hot air dryer (Blue Spark Systems SRL, Bucharest, Romania) at 60 °C, packed in polyethylene bags and stored in ambient conditions before analysis and use. Before being included in bird diets, they were shredded with a universal hammer mill (MCU 7.5 kW), with 1 mm mesh and analysed for the proximate composition (dry matter, ash, crude protein, crude fiber and ether extract) according to the standard methods of AOAC International [33] and carotenoid profiles. 

### 2.2. Experimental Design

All procedures related to birds’ care were performed according to the principles of the animal welfare stated by the Directive 2010/63/EU [34] regarding the protection of the animals used for experimental and other scientific purposes. The experimental procedures were approved by the Ethical Commission of National Research and Development Institute for Biology and Animal Nutrition (no. 52/30.07.2014). A four-week experiment was conducted on 168 Lohmann Brown layers (average weight = 1.953 ± 0.172 kg) at 43 weeks of age, assigned to four dietary treatments with 42 birds each (21 replicate/groups with 2 birds/pen). Birds where housed in an experimental hall equipped with Big Dutchman batteries (50 cm × 50 cm × 40 cm, with a floor slope of 12°) under controlled environmental conditions and monitored by a ViperTouch computer (temperature = 21.66 ± 1.71 °C, humidity = 57.78 ± 4.58%, ventilation = 2.41 ± 0.50% and light regimen = 16 h light/8 h darkness). The experiment was carried out during the period November–December. The feed was administrated once daily at 08:30 and water was available at all times. No vaccination treatment was applied to the birds throughout the experimental period (4 weeks).

### 2.3. Dietary Treatments

Diets formulations (Table 1) were calculated in agreement with the nutritional requirements of Lohmann Brown Layer hibrid [35]. The basal diet formulation was similar for all four groups. The control diet formulation (C) had a conventional structure (characterized by 2780 kcal metabolizable energy and 17.5% crude protein), commonly used by feed producers. Unlike the diet C, for the experimental groups (E1, E2 and E3) the diets included 6% flaxseed meal as a source of n-3 PUFA (characterized by 12.66% crude fat, of which 45.69 g α-linolenic acid (ALA)/100 g total FAME), supplemented with: 2% dehydrated kapia peppers (E1); 2% dehydrated sea buckthorn pomace (E2) or 2% dehydrated carrots (E3). The supplements have been added to experimental diets as rich sources of carotenoids (Table 2). The diets formulation was developed using a dedicated software (HYBRIMIN^®^ Futter 2008, Hybrimin GmbH & Co., Hessisch Oldendorf, Germany), in agreement with the feeding requirements of laying hens and all diets were isocaloric and isonitrogenous (Table 1). Throughout the feeding trial, a single batch of feed was manufactured and samples were collected (approx. 500 g/sample) in order to carry out the chemical analyses. After diets manufacturing, the feed was seated in bags, labelled and stored in a cool space until consumption. 

### 2.4. Performance Parameters

Body weight (g/hen), feed intake (g/day/hen), feed conversion ratio (g feed/g egg), laying percentage (%), egg size classification (%) and mortality were monitored throughout the experimental period. Body weight was measured at the beginning and the end of the experimental period. Feed intake and egg production were recorded daily. Eggs were collected and weighed every day, and egg production was expressed as average hen-day production, calculated from total eggs divided by the total number of hen-days. After daily weighing of all eggs, they were individually classified according to the European Council Directive (2006) [36] into four categories of eggs: extra-large (>73 g), large (73–63 g), medium (63–53 g) and small (<53 g), as reported by other researchers [37,38]. Data on feed intake and egg mass were used to calculate the feed conversion ratio (feed intake/egg mass; g/g). Intake of carotenoids (mg/day/hen) was calculated based on the content of carotenoids in the diet (mg/kg) × feed intake (g/day/hen). All performance parameters were determined for each replicate of treatment groups.

### 2.5. Sampling Collection and Procedures 

Eggs from each treatment were analysed after two respectively four weeks of feeding. One hundred and forty-four eggs (18 eggs/group/period) were collected randomly from each group and weighed individually for determination of the external and internal egg quality. Before the eggs were broken, eggshell thickness and eggshell breaking strength were measured using the Egg Shell Thickness Gauge (ORKA Technology LLC, Wanchai, Hong Kong) and the Egg Force Reader (ORKA Technology LLC), respectively. Then, the components of the eggs (albumen, yolk, shell) were manually separated and weighed using the same balance as for the whole egg. pH (albumen and yolk) was measured using a portable pH meter (Five Go F2-Food kit with LE 427IP67, Sensor Metler Tolledo, Greifensee, Switzerland) and Haugh unit score was determined using an Egg Analyzer TM (ORKA Technology LLC). After that, six yolk samples (3 eggs/sample) for each group were formed from the collected eggs (18/group/period) and assayed for yolk color, cholesterol content and fatty acids profile. The evolution of carotenoids in egg yolk was achieved by collecting 6 eggs/group after 0, 3, 9, 15, 21 and 27 days of feeding with dietary treatments. Yolk samples were stored at −20 °C until analysis. Before analysis, samples were allowed to achieve room temperature. In order to determine the fat oxidative stability after 28 days of eggs storage at 4 °C, 18 fresh yolk samples/group were analysed. 

Chemical analysis (fatty acids profiles, the oxidative stability of eggs, cholesterol and total fat content in yolk eggs) were determined on the dry yolk samples in the Chemistry laboratory at the National Research and Development Institute for Biology and Animal Nutrition, Balotesti (Romania). Carotenoid profiles and color measurement of yolk were determined on the fresh yolk samples in the laboratories of the Department of Horticulture & Food Science, University of Craiova, Craiova, (Romania).

### 2.6. Color Measurement 

Yolk color was measured according to CIELab color scale using a PCE-CSM1 reflectance colorimeter (PCE Instruments, Southampton, UK) calibrated against a white reference ceramic tile. Color was expressed as L* (lightness), a* (redness), and b* (yellowness) reflectance values of the CIELab system (Commission Internationale de l’Éclairaige). The analysis was performed on three samples from each group with four readings on each sample. The hue angle (h) was calculated as arctan (b*/a*) while chroma (C) was calculated as (a*2 + b*2)1/2. In addition, yolk color was determined every three days, on 5 eggs/group, by the Roche yolk color fan (Hoffman-La Roche Ltd., Basel, Switzerland; color scale from 15, dark orange, to 1, light pale).

### 2.7. Egg Yolk Carotenoid Analysis

Yolk samples were subjected to triplicate analyses for carotenoids using high-performance liquid chromatographic assay with diode-array detection at 450 nm, as described by Panaite et al. [10]. Standards of lutein, zeaxanthin, canthaxanthin, astaxanthin, lycopene, *β*-carotene and trans-*β*-apo-8′-carotenal were purchased from Sigma-Aldrich, Munich, Germany). Standard solutions were obtained by dissolving pure compounds in acetonitrile-methanol-ethyl acetate (60:20:20, *v*/*v*/*v*) containing butylated hydroxytoluene (BHT) (1% *w*/*v*). HPLC analyses were performed on a Finningan Surveyor Plus system (Thermo Electron Corporation, San Jose, CA, USA). Chromatographic separation was achieved by a reversed-phase Hypersil Gold C18 column (Thermo, Waltham, MA, USA) at 20 °C. Carotenoids were extracted from 0.5 g sample with 10 mL of petroleum ether:methanol:ethyl acetate (1:1:1, *v*/*v*/*v*) containing 0.1% BHT by homogenizing for 5 min at 2500 rpm using a Vortex homogenizer. The sample was centrifuged for 6 min at 6000 rpm and the supernatant was collected. The residue was extracted following the same procedure until the supernatant was colorless. The combined supernatants were washed by adding 10 mL of 5% NaCl solution, mixing vigorously and incubating for 30 min until two layers were separated. The upper layer was collected, evaporated to dryness under N2 flow and then re-dissolved in 2 mL of acetonitrile:methanol:ethyl acetate (60:20:20, *v*/*v*/*v*) containing BHT (1% *w*/*v*). The final solution was filtered through 0.45 µm membrane filters for HPLC injection. The mobile phase comprised acetonitrile:methanol (95:5, *v*/*v*) (A), acetonitrile:methanol:ethyl acetate (60:20:20, *v*/*v*/*v*) (B) and water (C). Carotenoids were eluted at a flow rate of 1.5 mL/min with the following gradient: 96% A and 4% C in the beginning, maintained for 10 min, changed linearly to 100% B in 13 min, maintained 5 min and returned to 96% A and 4% C in 2 min. Quantification was performed using Chrom Quest 4.2 software by comparing peak areas with those of the known standards.

### 2.8. Egg Yolk Fatty Acid Analysis and Lipid Quality Indices

Fatty acid profile from dried yolk samples (65 °C) was determined using a gas chromatograph Perkin-Elmer Clarus 500 (Shelton, MA, USA) according to the method previously described by Panaite et al. [10]. The fatty acids from yolk samples were converted to methyl esters of fatty acids (FAME) and their separation was performed on a DB-23 GC capillary column(60 m × 0.25 mm id × 0.25 µm), Agilent J&W GC Columns, USA using a flame ionization detector (FID). The results were expressed in g/100 g total fatty acid methyl esters (FAME). The average amount of each fatty acid was used to calculate the sum of the total saturated (SFA), total monounsaturated (MUFA) and total polyunsaturated (PUFA) fatty acids. 

### 2.9. Lipid Oxidative Status of the Yolk

The oxidative stability of the eggs was determined by estimating the primary lipid degradation products, as indicated by the peroxide values and the content of conjugated dienes and conjugated trienes, and the secondary lipid degradation products, as indicated by the levels of thiobarbituric acid reactive substances (TBARS), according to the methods described by Untea et al. [39], as follows:

Total lipids were extracted from eggs yolk using 5 g homogenized sample in 30 mL chloroform/methanol mixture (2:1, *v*/*v*). The homogenate was filtered in a separation funnel and 7.5 mL of a 0.88% KCl aqueous solution was added. The sample solution was left to rest for 20 h and the lower organic layer was collected and evaporated at room temperature to a constant weight. The peroxide value (PV) was determined by the ferric thiocyanate method and was expressed as milliequivalents of oxygen per kilogram of lipid (meq O2/kg). To the lipid extract sample (0.1 g), 9.9 mL of chloroform/methanol (7:3, *v*/*v*) solution was added. After vortexing, 50 μL of 10 mmol/L xylenol orange solution and 50 μL of FeCl2 solution (1000 mg/kg) were added and then the absorptivity at 560 nm was measured using a V-530 Jasco (Japan Servo Co. Ltd., Tokyo, Japan) spectrophotometer. Conjugated dienes (CD) and trienes (CT) were determined by UV spectrometry at 232 and 268 nm, respectively, and reported as specific absorbance of the lipid extract sample dissolved in 2,2,4-trimethylpentane (iso-octane). 

The TBARS values were calculated from a standard curve of malondialdehyde and expressed as milligrams of malondialdehyde (MDA) per kg of sample (mg MDA/kg). The sample (5 g) was mixed with 10 mL trichloroacetic acid (7.5%) and 5 mL butylated hydroxytoluene in ethanol (0.8%). The sample solution was centrifuged at 3000× *g* for 3 min. Aliquots of 2.5 mL were mixed with 1.5 mL of 0.8% aqueous thiobarbituric acid solution in a test tube and further incubated at 80 °C for 50 min. Following incubation, the sample was cooled under running water and the absorbance was read at 532 nm and corrected for unspecific turbidity after substraction from the value obtained at 600 nm.

### 2.10. Yolk Cholesterol Content

Cholesterol content of dried yolk samples (65 °C) was determined using a gas chromatograph Perkin-Elmer Clarus 500 (Shelton, MA, USA) according to AOAC Official Method 994.10: Cholesterol in foods [40]. The egg yolk samples were subjected to saponification using 50 mL of methanolic KOH solution (0.5 M) for one hour in a water bath. The next step is the extraction of cholesterol in petroleum ether, followed by concentration in a rotary evaporator and wash with distilled water to neutral pH. After removing the petroleum ether, the residue was taken up again with 3–5 mL of chloroform. Cholesterol was separated on a HP-5 GC capillary column (30 m × 0.32 mm id, 0.1 µm film thickness), Agilent J&W GC Columns, USA and detected on a flame ionization detector (FID). An 1 μL aliquot was injected in the GC column. Concentrations were calculated by comparing peak areas with those obtained from the standard solutions and were expressed as g cholesterol/100 g dried egg yolk. 

### 2.11. Statistical Analyses 

The data analyses were carried out using XLSTAT 2014 software (AddinSoft, Paris, France). The statistical effect of diets was tested by one way analysis of variance (ANOVA) at a significance level of 5%, followed by the Tukey comparison. Differences were considered significant when *p* < 0.05, and highly significant when *p* < 0.001. The effect of the studied factors (diet and time) on the external and internal egg quality, color, fatty acids profile was analysed by two-way ANOVA using XLSTAT 2014 (AddinSoft, Paris, France) followed by Tukey’s multiple range test. 

The following statistical model was used:Y_ij_ = µ + α_i_ + β_j_ + (αβ)_ij_ + ε_ijk_
where Y = the dependent variables, µ = the general mean, α_i_ and β_j_ = diet and time effects, (αβ)_ijk_ = the interaction between diet and time, and ε_ij_ = the random error. Differences were considered significant when *p* < 0.05, highly significant when *p* < 0.001, and 0.05 < *p*-value < 0.10 was discussed as tendencies.

## 3. Results

### 3.1. Proximate Composition, Fatty Acid and Carotenoid Profile of Dried Kapia Pepper, Sea Buckthorn Pomace and Carrot

Table 2 reveals a higher content of crude protein, crude fibre, and ether extract in sea buckthorn pomace compared with the other two vegetal sources. Regarding the content in polyunsaturated fatty acids, among the three sources studied, kapia pepper is characterized by a high total PUFA content, of which 21.5% are represented by n-3 PUFA (Table 2). Regarding the carotenoid profile, among the three sources, kapia pepper was the richest in astaxanthin, lutein, canthaxanthin and lycopene while the most abundant carotenoids recorded in sea buckthorn pomace were β-carotene and zeaxanthin. The richest source of β-carotene was carrot, its content being 4.30 times higher than in kapia pepper and 4.03 times higher than in sea buckthorn pomace. 

### 3.2. Performance Parameters

There was no difference (*p* > 0.05) in the final body weight, the laying percentage and the total number of eggs/period among the experimental groups (Table 3). Laying hens fed diets supplemented with linseed meal + sea buckthorn pomace and linseed meal + carrot had a significantly higher average daily feed intake than the hens fed the other two diets. At the same time, feed conversion ratio did not recorded differences between C, E1 and E3 groups. Hens fed diet supplemented with linseed meal + sea buckthorn pomace (E2) had a significantly higher feed conversion ratio than those fed conventional diet (C). The total carotenoid intake increased significantly as consequence of kapia pepper, sea buckthorn pomace and carrot supplementation in the diet. The highest carotenoid intake was observed in the group fed diet with addition of 2% carrot, followed by kapia pepper and sea buckthorn pomace addition. The highest percent of XL eggs was recorded in E3 group, significantly higher compared to C, E1, and E2. For L size classification, E2 group recorded a significantly higher percent than the other groups, while for the M size, the highest percent was recorded in E1 group. 

### 3.3. External and Internal Egg Quality Parameters

Table 4 shows the egg physical parameters recorded at 2 and 4 weeks of feeding. There were no differences (*p* > 0.05) between groups in the egg weight, albumen weight, yolk weight, egg shell weight, albumen pH and eggshell breaking strength (*p* > 0.05). The yolk pH was significantly affected by the diet (*p* < 0.0001) and feeding time (*p* < 0.0001). Thus, at 2 and 4 weeks of feeding, the yolk pH increased in the groups E1, E2, E3 than in the C group (*p* < 0.0001). The highest pH was recorded by the eggs yolk from E3 group. The pH value represents an important criteria in the quality of the eggs. According to Suresh et al. [41], pH of the fresh egg yolk is around 6.0 and increases to 6.4–6.9 during storage. In the present study, the yolk pH decreased with the time of feeding in E1 and E3, being higher at 4 than at 2 weeks of feeding. The eggs collected from hens fed C and E2 diets did not record significant differences at 4 weeks vs. 2 weeks of feeding. The Haugh unit was affected only by the time (*p* < 0.0001), being lower at 4 than at 2 weeks of feeding. 

Eggshell thickness was significantly influenced by diet (*p* = 0.0002) and time (*p* = 0.0306). At 2 weeks of feeding, the highest eggshell thickness was recorded in E2 group, significantly higher than C and E3. At 4 weeks of feeding, the highest eggshell thickness was recorded in E3 group. Regarding the effect of time on eggshell thickness, a higher value was recorded in E3 group at 4 weeks than at 2 weeks. This result showed that the supplementation with linseed meal + carrot in the hen’s diet for a longer period of time improved the eggshell thickness. 

### 3.4. Yolk Color and Carotenoid Concentration in Eggs Yolk

The results showed that adding dried kapia pepper, sea buckthorn pomace or carrots in the hen’s feed at a level of 2% had a favorable effect both on color (Table 5) and on the carotenoids’ accumulation in egg yolk (Figure 1).

The highest reddish color pigmentation of egg yolk was obtained from the group supplemented with 2% dried kapia peppers, followed by the groups supplemented with 2% dried sea-buckthorn pomace and 2% dried carrots. The enhanced color of egg yolk from the experimental groups is associated with the increased level of carotenoids in the feed, as a result of the dietary supplementation with these rich sources of carotenoids (Table 1). The Roche yolk color score increased significantly by the addition of dried kapia peppers in laying hens diet both at weeks 2 (*p* < 0.05) and 4 (*p* < 0.05) of the experiment compared with the control (Table 5). 

Figure 1 presents the evolution of carotenoids concentration in the fresh yolk samples during the feeding period. Generally, the maximum carotenoids content accumulated in yolk after 3 weeks from the start of the experiment. After 3 weeks of feeding, the contents of astaxanthin (Appendix A), lutein (Appendix A), zeaxanthin (Appendix A), canthaxanthin (Appendix A), beta-carotene (Appendix A), and total carotenoids (Appendix A), of the egg yolk from the group supplemented with dried kapia pepper (E1) were significantly (*p* < 0.05) higher as compared to the control diet. The sea buckthorn meal treatment increased (*p* < 0.05) the yolk content of astaxanthin, zeaxanthin and canthaxanthin, as compared with the control, whereas lutein and beta-carotene did not recorded any differences. Beta-carotene concentration was significantly (*p* < 0.05) higher after 3 weeks of feeding in the yolk of the carrot treatment compared to control treatment, whereas the astaxanthin content was significantly (*p* < 0.05) lower. Overall, carotenoid sources included in the experimental diets were positively influenced the total carotenoid content in the egg yolks. The egg yolk Roche color score was significantly increased by the addition of dried sea-buckthorn pomace and carrots in laying hens diets as compared with the control group but no significant differences were found between the egg yolk color scores of sea-buckthorn pomace and carrots supplemented groups. Addition of dried sea buckthorn pomace in laying hens diet resulted also in an increased reddish color pigmentation of egg yolk. Yolk color reached a score of 7.50 in the E2 treatment group as compared with 5.94 in the control group after 2 weeks of feeding.

Results of the CIELab analysis showed that feeding laying hens with a diet supplemented with 2% dried sea buckthorn pomace during 4 weeks determined a significant increase of yolk redness (a* values, from −1.32 to 4.41), yellowness (b* values, from 39.09 to 49.21) and Chroma (from 40.40 to 49.41) as compared with the control group, while L* values were not significantly affected. Similar trends have been reported by several previous studies [32,42]. After 3 weeks of feeding, yolks from the group supplemented with dried sea buckthorn pomace showed a significantly higher content of total carotenoids deposition (17.613 mg/kg) as compared with those from the control group (15.385 mg/kg). This was mainly due to the increment of zeaxanthin and cantaxanthin content of the yolk as a result of the dietary supplementation with dried sea buckthorn pomace (Figure 1).

Although carrot supplementation led to a higher carotenoid content of the yolk as compared to the control diet, the effect on the yolk color was reduced. The carrot treatment increased the yolk content of zeaxanthin (>1.5-fold) and β-carotene (>6-fold) after 3 weeks of feeding as compared with the control (Figure 1). The lutein content increased also but the increment was not significant (*p* < 0.05). Addition of 2% dried carrots resulted in a significant increase of egg yolk yellowness (b* values) and a slight decrease of lightness (L* values) as compared to the control group. However, no significant differences were found between the egg yolk redness (a*) of control and carrots supplemented groups both after 2 and 4 weeks of feeding. Similar increases of yolk yellowness (b* values) have been previously reported as a result of a 2 weeks supplementation of laying hens diet with different colored carrot varieties. However, unlike the results of this study, yolks from hens fed carrots were significantly darker (lower L* values) and more reddish (higher a* values) than the control egg yolks, especially when using purple carrots [43,44].

### 3.5. Egg Yolk Fatty Acids Profile

The data on egg yolk fatty acids profile are summarized in Table 6. The dietary supplementation with linseed meal and carotenoid sources in laying hen diets significantly (*p* < 0.0001) improved the alpha-linolenic fatty acid (ALA) content in all three experimental groups, both after 2 and 4 weeks, compared to the control. The highest increase of ALA content in diet and time was recorded in E2 group with 6% linseed meal and 2% dried sea buckthorn pomace. Regarding the docosapentaenoic (DPA) and docosahexaenoic acids (DHA), the largest increase (*p* < 0.0001) was obtained at 2 weeks in the eggs of E1 group which included linseed meal and 2% dehydrated kapia pepper in the diet. 

Egg yolk analysis after 4 weeks of feeding highlighted significantly higher DPA and DHA concentrations in the group with 6% linseed meal and 2% dehydrated carrot (E3), compared to the control group. Overall, diet and time together did not influence (*p* < 0.05) the egg yolk n-3 PUFA fatty acids (Table 7). 

Compared to the control group, the carrot and sea buckthorn pomace added to the diets resulted in the lowest SFA concentrations. Although the total PUFA concentration decreased in experimental groups, the n-3 PUFA content was significantly (*p* < 0.0001) improved compared to the control. After 2 weeks, the E1 egg yolk had the highest concentration, while after 4 weeks the tendency changed and the eggs with the highest concentration of n-3 PUFA were those from E2 group. This led to an improvement (*p* < 0.0001) in the ratio of n-6/n-3 PUFA fatty acids in all three experimental groups, especially in the group with linseed meal and dried sea buckthorn pomace (E2) as compared to control (C). 

### 3.6. Egg Yolk Cholesterol Content and Lipid Oxidation

The cholesterol content and lipid oxidation status of the egg yolk from hens fed with tested feeds are given in Table 8. As can be seen, although the egg yolk from the experimental diets had a higher fat content as compared to the control, there was a decrease in their cholesterol content. An improvement was also observed regarding the lipid oxidative status of egg yolk. Among the primary lipid oxidation parameters, the peroxide value was significantly (*p* < 0.05) lower in all experimental groups as compared with control. The concentrations of conjugated dienes did not record any differences (*p* > 0.05) between groups. Compared to the control, carrots have shown the best efficiency (*p* < 0.05) in slowing down lipid degradation processes. TBARS values decreased significantly (*p* < 0.05) in the groups supplemented with kapia pepper and carrot. 

## 4. Discussion

### 4.1. Chemical Composition 

The results on chemical composition of carotenoid sources used in this study revealed that the richest source of carotenoids is carrot, the most abundant carotenoid in carrot being β-carotene. The results are consistent with data reported in previous studies [45,46]. Koka and Karadeniz [47] showed that the β-carotene content of carrot ranged from 29 to 130 mg/kg. The other two are also rich sources of carotenoids, the major carotenoids in sea buckthorn pomace being β-carotene and zeaxanthin while the carotenoid profile of kapia pepper is dominated by β-carotene, lutein and zeaxanthin. Similar results were published previously [30,48,49,50]. 

### 4.2. Performance Parameters

In the present study, the final body weight, laying percentage and the total number of eggs/period did not record any differences among the experimental groups (Table 3). Li et al. [51] reported similar results when using 0.3, 0.6, 1.2, 2.0, 4.8 or 9.6 mg/kg red pepper pigment in laying hens diet. 

Results of this study show an increase in the feed intake for laying hens fed diets supplemented with linseed meal + sea buckthorn pomace (E2) and those with linseed meal + carrots (E3) than control (C) and those supplemented with linseed meal + kapia pepper (E1). Feed conversion ratio was significantly (*p* < 0.05) lower in C group than in E2 group, but insignificantly lower (*p* > 0.05) as compared with E1 and E3 group. Lokaewmanee et al. [24] showed that the addition of 0.5% red pepper to the hen’s diet did not significantly influence the feed intake, final body weight, egg production and feed efficiency. Contrarily, Abou-Elkhair et al. [52] showed that the dietary inclusion of 5 g/kg red pepper improved (*p* < 0.05) egg production, feed conversion ratio compared with control. By replacing 5% of the feed wheat with sea buckthorn fruit residues (berries resulting after extracting the juice and then dried), Shaker et al. [32] did not record any difference in the hen’s performance. 

### 4.3. External and Internal Egg Quality Parameters 

Results from the present study showed that dietary supplementation with linseed meal + kapia pepper, linseed meal + sea buckthorn pomace or linseed meal + carrot did not affect the egg weight, albumen weight, yolk weight, egg shell weight, albumen pH and eggshell breaking strength (*p* > 0.05). Similar results were obtained by Lokaewmanee et al. [24] when fed diets supplemented with red pepper. The authors reported no differences in egg weight, shell-breaking strength, shell thickness and Haugh units. Shaker et al. [32] showed that replacement of 5% of the feed wheat with sea buckthorn fruit residues did not affect egg quality. On the contrary, during 32 to 40 weeks of age, dietary inclusion of 5 g/kg hot red pepper in the hen’s diet increased the egg weight compared with control group [52]. Chand et al. [53] showed that sea buckthorn seed supplementation (2 and 3 g/kg of feed) increased egg weight (week 39 and 40). 

In the present study, the yolk pH decreased with the time of feeding in the experimental groups. This result suggests that although the diets were enriched in n-3 PUFA, the addition of carotenoids sources delayed the oxidative processes.

Regarding the effect of time on eggshell thickness, an increase was observed in E3 group at 4 weeks than at 2 weeks. This achievement highlighted that the supplementation of linseed meal + carrot in the hen’s diet for a longer period of time improved the eggshell thickness. 

### 4.4. Yolk Color and Carotenoid Content

The Roche Color Fan score of the yolk from the kapia pepper treatment (E1 group) reached the overall significantly highest level (15.00) after 4 weeks of feeding as compared with the other treatment groups. Improvements in egg yolk color after supplementation of hens diet with sweet or hot red pepper have been previously reported in many studies [52,54,55] and they have been attributed to the high content of carotenoid pigments in red pepper. The colorimetric measurements revealed that the addition of kapia pepper in laying hens diet resulted in a significant (*p* < 0.05) increase of the yolk redness (a* values) and yellowness (b* values) while the yolk lightness (L* values) significantly (*p* < 0.05) decreased. Some other studies found also that the yolks from hens fed a diet supplemented with red pepper powder were redder, yellower and darker than those from the hens fed the basal diet [24,51,56]. The greatest increment was observed in parameter a* for the red color as a result of the dietary supplementation with kapia pepper (from −1.96 in the control yolk samples to 18.42 in the E1 yolk samples after only 2 weeks of feeding). Similar results have been reported by Hamilton et al. [56] who found that yolk redness (a* values) correlated highly with zeaxanthin, capsanthin and total carotenoids content in eggs from hens supplemented with red pepper oleoresin. It is well known that the yolk color and its components (redness and yellowness) are determined by the total carotenoid content and by the ratio between the yellow carotenoids (lutein, zeaxanthin) and the red carotenoids (canthaxanthin, astaxanthin and capsanthin). 

Results from the present study revealed lutein and zeaxanthin as main carotenoids in the yolk of the control samples. Supplementation of carotenoids in laying hens diet from all the three different dietary sources resulted in the accumulation of zeaxanthin in a greater quantity than lutein. A similar observation was made by Dumbrava et al. [31] after adding sea buckthorn berry flour in laying hens diet. The addition of dried kapia pepper, sea buckthorn pomace or carrots in the hen’s feed had a favorable effect on the carotenoids’ accumulation in egg yolk even after 3 days of feeding. As we expected, the higher concentration of beta-carotene was recorded in the yolk of the group supplemented with carrot (E3). This result is due to the high concentration of beta- carotene in the diet as consequence of its carrot supplementation. Similar finding was described by Hammershøj et al. [44] when administered 70 g day/1 per hen of different carrot varieties (Purple Haze and Rainbow). The well-known health benefits of carotenoid intake (e.g., reduced prevalence of cardiovascular diseases, diabetes, cancer, etc.) have previously been attributed mainly to their antioxidant properties (e.g., radical quenching), anti-inflammatory effects [57,58].The weak coloring effect of carrots found in this study was due to the low β-carotene content of the carrots used in this study (32.58 mg/kg, Table 2) and to the low yolk pigmentation efficiency of β-carotene, which has been attributed to its lower polarity as compared to other carotenoids [45,59].

### 4.5. Egg Yolk Fatty Acids Profile

The concept of enriching eggs in n-3 PUFA through diet supplementation with oil seeds or oils rich in n-3 PUFA is not an innovative one. Although several studies have been published since the last century on the effects of feeding flaxseed meal or flaxseed oil on the performance of eggs and the fatty acid composition of eggs [60,61], this topic is still in the attention of researchers. In the present study, the supplementation of hens diet with carotenoid sources, along with linseed meal, led to egg fatty acid profile improvement, especially the n-3 PUFA content. This way of enriching eggs in fatty acids and bioactive compounds using natural products is one of the methods currently used to biofortify eggs [1]. The composition of the egg in fatty acids is closely related to fatty acids from feed. As expected, because the feeds from E1 and E2 group had the highest n-3 PUFA concentrations, after 2 weeks, eggs from these groups had also the highest level of n-3 PUFA. After 4 weeks, egg yolk from E2 group continued to have the highest level of n-3 PUFA alongside the eggs from E3 group. It was also observed that the new diet formulations led to a much lower value (*p* < 0.0001) of the n-6/n-3 PUFA ratio. This fact was most evident in the diet that included sea buckthorn pomace, well known for its powerful antioxidant properties [62] and its capacity to improve the efficiency and the oxidative stability of feed [32]. On the other hand, the inclusion of kapia pepper in laying hen diets significantly increased the content of docosapentaenoic acid (DPA, C22:5n-3) and docosahexaenoic acid (DHA, C22:6n-3). DPA and DHA are essential fatty acids that play an important role in human health by reducing the risk of chronic diseases [63]. The recommended daily intake for DHA is between 300 and 500 mg/day [64]. Thus, supplementing hen diets with natural sources of fatty acids is a good solution to increase n-3 PUFA content of eggs. In contrast to the results found in this study, Omri et al. [1] showed that inclusion of linseed meal (4.5%) together with tomato (1%) and sweet pepper (1%) to hen diets did not significantly influence the egg yolk DHA content. Enriching n-3 PUFA content of egg yolk continues to remain a subject of major interest for both researchers in the field of poultry and medicine. The n-3 PUFA possesses a crucial role in human nutrition, being involved in several physiological processes such as regulation of inflammation, alteration of gene expression, modification of membrane raft structure and function [65], and prevent coronary disease, arrhythmias, heart failure (HF), and dyslipidemia [66].

### 4.6. Cholesterol Content and Lipid Oxidative Status of the Yolk

The literature contains information regarding the high level of egg cholesterol (183–386 mg/egg) from hens fed with standard diets [1]. In the current work, linseed meal was used as raw material rich in polyunsaturated fatty acids in the experimental treatments, together with different sources of carotenoids with antioxidant properties. This led to a decrease in egg yolk cholesterol level, compared to the control group fed with a conventional diet. The highest (*p* < 0.05) decrease in cholesterol level was recorded for eggs from the experimental group with 6% linseed meal and 2% dried sea buckthorn pomace. This finding was in agreement with that of Chand et al. [53] who reported that the dietary inclusion of sea buckthorn seed powder at a rate of 3 g/kg of feed improved significantly (*p* < 0.01) the egg yolk cholesterol level (14.45 ± 0.42 mg/g of yolk). A decrease with 11.44% in egg yolk cholesterol levels was found in carrot supplemented group. Carrot is a rich source of antioxidant flavonoids [67], which positively influenced cholesterol content along with linseed meal.

It is well-known that increasing polyunsaturated fatty acids concentration in eggs favours the lipid oxidation processes. In this study, although the eggs obtained from experimental groups were enriched in fatty acids (from flaxseed meal), the degradation products (peroxides, conjugated dienes, conjugated trienes and malondialdehyde) were recorded in a lower or similar concentration with the control group. Compared with the eggs collected from the control group, those collected from the group fed PUFA—enriched diet and 2% dehydrated carrot had the lowest peroxide value, with 78.30% lower, respectively. This may be due to the highest content of carotenoids of the carrots. An inverse proportionality between carotenoids concentration and primary degradation products has been reported previously [10]. The concentration of conjugated trienes were lower (*p* < 0.05) in the yolk obtained from groups including sea buckthorn pomace and carrot in the diet. It is difficult to compare this finding with those of other studies because similar literature works are lacking. Carotenoid-enriched eggs have been associated with a decrease in MDA content in egg yolks [68,69] due to the role of carotenoids as natural antioxidants [70]. Similarly, in the present study, a significantly lower MDA content in yolk was observed in both the pepper and carrot-fed groups compared to the control group. Akdemir et al. [71] found that dietary supplementation with tomato powder (5 or 10 g of tomato powder per kg of diet) led to an increase in the concentration of carotenoids and vitamin A in egg yolk and, as a result, reduced the lipid peroxidation of the yolk. Similarly, An et al. [72] showed that dietary lycopene as carotenoid source reduced the MDA level in serum and eggs stored at 24℃ for four weeks. The literature is rather scarce in information on the effect of carotenoid-rich supplements included in hens diet enriched in polyunsaturated fatty acids on the oxidative degradation of yolk lipids. 

It is noteworthy that, although a lower concentration of total carotenoids was detected, kapia pepper was as effective as carrot in decreasing the MDA content in eggs. This effect could be probably attributed to the presence of additional natural antioxidants in kapia pepper such as capsaicinoids [73] or ascorbic acid [74], which could potentiate its antioxidant properties and protect against peroxidation of yolk lipids. In general, the synergism between fat-soluble antioxidants such as α-tocopherol and carotenoids, and water-soluble ones such as polyphenols and vitamin C in preventing the oxidation of human serum in vitro has been confirmed [75], but in vivo studies to certify these results are lacking. 

## 5. Conclusions

The results of the present study showed that kapia pepper, sea buckthorn pomace and carrot included in linseed enriched laying hens’ diets increased the feed intake and total carotenoids intake, this resulting in an increase in carotenoids’ accumulation in egg yolk. The carotenoids sources included in linseed enriched laying hens’ diets improved n-3 fatty acids contents with more health-promoting n-6/n-3 ratio, and positively affected the physical properties (e.g., yolk pH, egg thickness), color, cholesterol content and oxidative stability. Kapia pepper supplementation was most successful in increasing the egg yolk color of laying hens. 

In conclusion, dietary supplements offer a higher functionality of eggs and respect the consumer’s acceptability. A prolonged feeding period negatively affected the yolk pH, eggshell breaking strength, yolk color, n-6, n-6/n-3, but positively affects L*, a*and ΣPUFA. Further research must be conducted to investigate the extra benefits of these dietary carotenoids’ sources.

## Figures and Tables

**Figure 1 foods-10-01246-f001:**
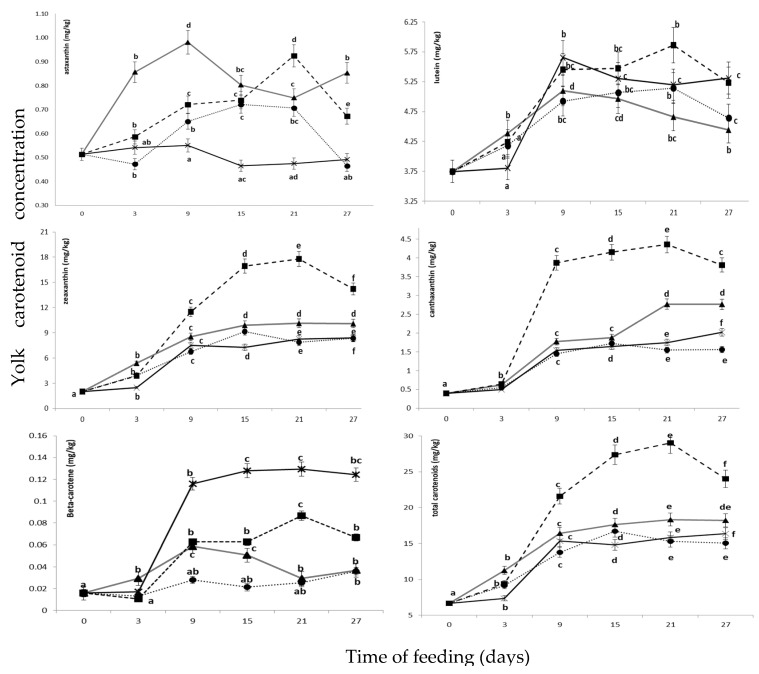
Time evolution (0–27 days) of yolk carotenoid concentration according to the dietary treatments: control (●), experimental E1 (■), experimental E2 (▲) and experimental E3 (**×**). C—control diet; E1—diet supplemented with 6% linseed meal + 2% dried kapia pepper; E2—diet supplemented with 6% linseed meal + 2% dried sea buckthorn pomace; E3—diet supplemented with 6% linseed meal + 2% dried carrot. ^a–f^ Mean values within a line not sharing the same superscripts are significantly different at *p* < 0.05.

**Table 1 foods-10-01246-t001:** Ingredients and chemical composition of the diets (% as fed).

Items	Experimentl Diets ^1^			
	Control (C)	Kapia Pepper (E1)	Sea Buckthorn Pomace (E2)	Carrot (E3)
Ingredients, %				
Corn	50.00	50.00	50.00	50.00
Soybean meal	29.52	20.57	19.91	21.30
Sunflower meal	4.22	7.21	8.00	6.49
Vegetable oil	3.50	1.31	1.16	1.3
Linseed meal	-	6.00	6.00	6.00
Kapia pepper (dehydrated)	-	2.00	-	-
Sea buckthorn pomace (dehydrated)	-	-	2.00	-
Carrot (dehydrated)	-	-	-	2.00
DL-methionine	0.08	0.17	0.17	0.17
L-lysine	0.02	0.02	0.04	0.02
Calcium carbonate	9.81	9.85	9.85	9.85
Monocalcium phosphate	1.40	1.42	1.42	1.42
Salt	0.40	0.40	0.40	0.40
Choline premix	0.05	0.05	0.05	0.05
Vitamin-mineral premix ^2^	1.00	1.00	1.00	1.00
Calculated analysis, %				
Metabolizable energy (MJ/kg)	2780.00	2780.00	2780.00	2780.00
Crude protein	17.50	17.10	17.20	17.10
Crude fat	4.20	4.70	4.83	4.69
Chemical analysis, %				
Fatty acids composition (% of total fat)				
ΣSFA	28.81	22.30	25.02	25.82
ΣMUFA	34.82	29.79	32.93	34.04
∑ PUFA, from which:	35.91	47.60	41.89	39.83
Σ n-3 PUFA	2.67	10.63	9.29	7.84
∑ n-6 PUFA	33.24	36.98	32.60	31.99
∑ n-6/∑ n-3	12.45	3.48	3.51	4.08
Carotenoid profile (mg/kg)				
Astaxanthin	0.122	0.129	0.121	0.124
Lutein	0.309	0.429	0.329	0.314
Zeaxanthin	0.333	0.385	0.465	0.323
Canthaxanthin	0.125	0.154	0.127	0.128
β-carotene	0.056	0.218	0.231	0.719
Total carotenoid, mg/kg	0.945	1.315	1.273	1.608

^1^ C—Control diet; E1—diet supplemented with 6% linseed meal +2% dried kapia pepper; E2—diet supplemented with 6% linseed meal +2% dried sea buckthorn pomace; E3—diet supplemented with 6% linseed meal + 2% dried carrot; ΣSFA: sum of saturated fatty acids; ΣPUFA: sum of polyunsaturated fatty acids; Σ n-3 PUFA = C18:3n-3 + C18:4n-3; Σ n-6 PUFA = C18:2n-6 + C20:2n-6 + C20:3n-6 + C20:4n-6; n.d.= below detection level; “-” = no specific ingredient included. ^2^ Vitamin mineral premix added at 1% to the diet contained (per kg feed): 1,350,000 IU/kg vitamin A; 300,000 IU/kg vitamin D3; 2700 IU/kg vitamin E; 200 mg/kg vitamin K; 200 mg/kg vitamin B1; 480 mg/kg vitamin B2; 1485 mg/kg pantothenic acid; 2700 mg/kg nicotinic acid; 300 mg/kg vitamin B6; 4 mg/kg vitamin B7; 100 mg/kg vitamin B9; 1.8 mg/kg vitamin B12; 2500 mg/kg vitamin C; 7190 mg/kg manganese; 6000 mg/kg iron; 600 mg/kg copper; 6000 mg/kg zinc; 50 mg/kg cobalt; 114 mg/kg iodine; 18 mg/kg selenium.

**Table 2 foods-10-01246-t002:** Proximate composition, fatty acid and carotenoid profile of dried kapia pepper, sea buckthorn pomace and carrot.

Nutrients	Kapia Pepper	Buckthorn Pomace	Carrot
*Proximate composition (%)*			
Dry matter	91.31 ± 3.78	94.60 ± 4.42	88.29 ± 3.19
Crude protein	9.34 ± 0.35	15.42 ± 0.68	6.11 ± 0.28
Ether extract	1.34 ± 0.08	15.08 ± 0.56	0.92 ± 0.04
Crude fibre	9.15 ± 0.38	30.84 ± 1.23	7.03 ± 0.44
Ash	5.38 ± 0.28	1.94 ± 0.12	6.27 ± 0.26
*Fatty acids composition (% of total fat)*			
ΣSFA	27.72 ± 1.21	34.28 ± 1.57	30.09 ± 1.39
ΣMUFA	14.13 ± 0.68	57.10 ± 2.66	23.42 ± 1.07
∑PUFA, from which:	58.04 ± 2.78	8.31 ± 0.46	46.28 ± 1.89
Σ n-3 PUFA	12.48 ± 0.45	2.74 ± 0.22	2.99 ± 0.18
∑ n-6 PUFA	45.56 ± 2.18	5.57 ± 0.25	43.29 ± 1.82
∑ n-6/∑n-3	3.65 ± 0.23	2.04 ± 0.14	14.50 ± 0.65
*Carotenoid profile (mg/kg)*			
Astaxanthin	0.501 ± 0.04	0.478 ± 0.03	0.147 ± 0.01
Lutein	5.631 ± 0.29	0.696 ± 0.05	0.237 ± 0.02
Zeaxanthin	1.528 ± 0.07	6.960 ± 0.36	0.031 ± 0.01
Canthaxanthin	0.971 ± 0.04	0.136 ± 0.01	n.d.
Trans-apo-carotenal	0.161 ± 0.01	0.236 ± 0.02	0.002 ± 0.001
Lycopene	0.351 ± 0.02	0.091 ± 0.01	0.259 ± 0.01
β-carotene	7.576 ± 0.34	8.076 ± 0.38	32.586 ± 1.46
Total carotenoid content	16.719 ± 0.75	16.673 ± 0.82	33.262 ± 1.51

n.d. = not detected.

**Table 3 foods-10-01246-t003:** Performances of laying hens fed with different sources of carotenoids (average values/group).

Items	Dietary Treatments				SEM	*p*-Value
	C	E1	E2	E3		
Initial body weight (g/hen)	1951.00	1950.50	1958.57	1952.50	19.118	0.9988
Final body weight (g/hen)	1940.00	1968.42	1961.58	1930.53	19.943	0.8941
Feed intake (g/day/hen)	115.82 ^b^	115.96 ^b^	119.53 ^a^	120.96 ^a^	0.656	0.0075
Feed conversion ratio (g feed/g egg)	2.02 ^b^	2.07 ^ab^	2.13 ^a^	2.10 ^ab^	0.015	0.0670 *
Total carotenoid intake (mg/day/hen)	0.11 ^c^	0.15 ^b^	0.15 ^b^	0.19 ^a^	0.006	<0.0001
Laying percentage (%)	91.51	91.75	90.89	92.70	0.480	0.6139
Total eggs/period (eggs)	1150	1156	1134	1169		
Egg size classification **						
Extra-large egg (XL), >73 g (%)	4.29 ^b^	3.41 ^b^	4.35 ^b^	6.25 ^a^	0.261	0.0003
Large egg (L), 63–73 g (%)	50.57 ^b^	38.44 ^a^	57.25 ^c^	49.43 ^b^	0.862	<0.0001
Medium egg (M), 53–63 g (%)	45.65 ^b^	56.62 ^a^	39.25 ^c^	43.93 ^b^	0.831	<0.0001
Small egg (S), <53 g (%)	2.65 ^b^	5.13 ^a^	2.71 ^b^	3.28 ^b^	0.273	0.0005

C—control diet; E1—diet supplemented with 6% linseed meal + 2% dried kapia pepper; E2—diet supplemented with 6% linseed meal + 2% dried sea buckthorn pomace; E3—diet supplemented with 6% linseed meal + 2% dried carrot; SEM, standard error of the mean. * ANOVA value of p was not significant, the superscripts are declared according to the Tuckey test results; ** European Council Directive (2006) [36]. ^a–c^ Mean values within a row not sharing the same superscripts are significantly different at *p* < 0.05.

**Table 4 foods-10-01246-t004:** External and internal quality parameters of the eggs.

	Egg Weight and Components (g)				Fresh Egg (Value)			Shell Quality (mm; kgF)	
	Whole Egg	Albumen	Yolk	Eggshell	Albumen pH	Yolk pH	Haugh Unit	Eggshell Thickness	Eggshell Breaking Strength
2 weeks									
C	62.67	38.26	16.02	8.39	8.29	5.79 ^c^	74.60 ^a^	0.34 ^bc^	3.96
E1	60.82	36.69	15.7	8.43	8.29	5.97 ^a^	75.92 ^a^	0.34 ^abc^	4.27
E2	61.02	36.84	15.88	8.3	8.38	5.92 ^b^	75.82 ^a^	0.36 ^ab^	4.30
E3	60.93	36.36	15.94	8.62	8.29	6.00 ^a^	76.13 ^a^	0.34 ^bc^	3.90
4 weeks									
C	61.09	37.00	15.84	8.25	8.41	5.81 ^c^	62.22 ^b^	0.33 ^c^	3.90
E1	61.56	37.17	15.72	8.59	8.27	5.90 ^b^	65.59 ^b^	0.36 ^ab^	3.98
E2	62.55	37.36	16.50	8.66	8.30	5.90 ^b^	63.13 ^b^	0.36 ^ab^	3.95
E3	60.80	36.47	15.56	8.51	8.30	5.90 ^b^	61.87 ^b^	0.37 ^a^	3.63
SEM	0.565	0.545	0.294	0.164	16.139	0.006	1.981	0.006	0.167
Main effects									
Diet									
C	61.88	37.63	15.93	8.32	8.35	5.80 ^c^	68.41	0.33 ^b^	3.93
E1	61.19	36.93	15.71	8.51	8.28	5.94 ^a^	70.75	0.35 ^a^	4.13
E2	61.78	37.10	16.19	8.48	8.34	5.91 ^b^	69.47	0.35 ^a^	4.12
E3	60.87	36.42	15.75	8.56	8.30	5.95 ^a^	69.00	0.36 ^a^	3.77
Time									
2 weeks	61.36	37.04	15.89	8.43	8.31	5.92 ^a^	75.62 ^a^	0.345 ^b^	4.109 ^a^
4 weeks	61.50	37.00	15.91	8.50	19.75	5.88 ^b^	63.20 ^b^	0.354 ^a^	3.867 ^b^
*p*-Value									
Diet	0.2255	0.1732	0.3477	0.488	0.3959	<0.0001	0.678	0.0002	0.0998
Time	0.7232	0.9232	0.9234	0.5552	0.3182	<0.0001	<0.0001	0.0306	0.0432
Diet × Time	0.0436	0.3269	0.3663	0.3631	0.3947	<0.0001	0.8003	0.0083	0.8224

C—control diet; E1—diet supplemented with 6% linseed meal + 2% dried kapia pepper; E2—diet supplemented with 6% linseed meal + 2% dried sea buckthorn pomace; E3—diet supplemented with 6% linseed meal + 2% dried carrot; SEM, standard error of the mean. ^a–c^ Mean values within a row not sharing the same superscripts are significantly different at *p* < 0.05.

**Table 5 foods-10-01246-t005:** Roche color fan score and Hunter color parameters of the eggs yolk.

Parameter	Yolk Roche Color Fan Score	L*	a*	b*	C	h
2 weeks						
C	5.94 ^c^	71.87 ^ab^	−1.96 ^d^	40.36 ^b^	39.16 ^c^	93.14 ^a^
E1	14.94 ^a^	61.577 ^c^	18.42 ^a^	46.86 ^ab^	50.35 ^a^	68.56 ^d^
E2	7.50 ^b^	69.34 ^ab^	2.39 ^c^	45.59 ^ab^	45.68 ^abc^	87.17 ^bc^
E3	7.50 ^b^	75.25 ^a^	−1.92 ^d^	43.24 ^ab^	43.27 ^abc^	91.34 ^a^
4 weeks						
C	5.89 ^c^	69.81 ^ab^	−1.32 ^d^	39.09 ^b^	40.40 ^c^	91.36 ^a^
E1	15.00 ^a^	61.20 ^c^	18.33 ^a^	46.17 ^ab^	49.68 ^ab^	68.36 ^d^
E2	6.22 ^c^	69.22 ^ab^	4.41 ^b^	49.21 ^a^	49.41 ^ab^	84.91 ^c^
E3	6.33 ^c^	66.78 ^bc^	−1.40 ^d^	41.68 ^ab^	41.74 ^bc^	90.16 ^ab^
SEM	0.156	1.484	0.409	1.853	1.8721	0.718
Main effects						
Diet						
C	5.92 ^c^	70.84 ^a^	−1.64 ^c^	39.73 ^b^	39.78 ^b^	92.25 ^a^
E1	14.97 ^a^	61.39 ^b^	18.37 ^a^	46.51 ^a^	50.02 ^a^	68.46 ^c^
E2	6.92 ^b^	69.28 ^a^	3.40 ^b^	47.40 ^a^	47.54 ^a^	86.04 ^b^
E3	6.86 ^b^	71.02 ^a^	−1.66 ^c^	42.46 ^ab^	42.51 ^b^	90.75 ^a^
Time						
2 weeks	8.97 ^a^	69.51 ^a^	4.23 ^b^	43.70	44.62	85.05 ^a^
4 weeks	8.36 ^b^	66.75 ^b^	5.00 ^a^	44.36	45.31	83.70 ^b^
*p*-Value						
Diet	<0.0001	<0.0001	<0.0001	0.0004	<0.0001	<0.0001
Time	<0.0001	0.0122	0.0106	0.6167	0.603	0.0108
Diet × Time	<0.0001	0.0249	0.0839	0.5176	0.5125	0.5212

C—control diet; E1—diet supplemented with 6% linseed meal + 2% dried kapia pepper; E2—diet supplemented with 6% linseed meal + 2% dried sea buckthorn pomace; E3—diet supplemented with 6% linseed meal + 2% dried carrot; L* (lightness), a* (redness), b* (yellowness), h (hue angle calculated as b*/a*), C (chroma) calculated as (a*2 + b*2)1/2. SEM, standard error of the mean; ^a–d^ Mean values within a row not sharing the same superscripts are significantly different at *p* < 0.05.

**Table 6 foods-10-01246-t006:** Effects of using linseed meal and carotenoid sources on egg yolk fatty acids profile (g acid/100 g total FAME).

Item	SFA			MUFA			PUFA					
							n-6			n-3		
	C14:0	C16:0	C18:0	C14:1	C16:1	C18:1	C18:2n6	C20:2n6	C20:4n6	C18:3n3	C22:5n3	C22:6n3
2 weeks												
C	0.32 ^a^	23.36 ^ab^	11.00 ^bc^	0.05 ^abc^	2.77 ^cd^	33.04 ^a^	20.34 ^a^	0.13	3.86 ^abc^	0.80 ^c^	0.14 ^b^	2.15 ^b^
E1	0.25 ^bc^	22.76 ^b^	11.05 ^bc^	0.06 ^ab^	3.04 ^bcd^	33.81 ^a^	17.79 ^b^	0.13	3.62 ^bc^	1.68 ^ab^	0.27 ^a^	3.44 ^a^
E2	0.25 ^bc^	23.19 ^ab^	11.16 ^abc^	0.05 ^ab^	3.59 ^a^	34.16 ^a^	17.02 ^b^	0.12	3.48 ^bc^	1.82 ^a^	0.20 ^ab^	3.17 ^a^
E3	0.27 ^b^	23.17 ^ab^	10.55 ^c^	0.06 ^a^	3.51 ^ab^	34.26 ^a^	17.97 ^b^	0.14	3.33 ^c^	1.71 ^ab^	0.24 ^a^	3.12 ^a^
4 weeks												
C	0.27 ^b^	23.62 ^ab^	12.53 ^a^	0.04 ^c^	2.15 ^e^	30.25 ^b^	21.15 ^a^	0.12	4.31 ^a^	0.74 ^c^	0.14 ^b^	2.32 ^b^
E1	0.25 ^bc^	23.72 ^a^	11.67 ^abc^	0.05 ^abc^	2.87 ^cd^	33.24 ^a^	17.79 ^b^	0.09	3.65 ^bc^	1.55 ^ab^	0.26 ^a^	3.20 ^a^
E2	0.24 ^bc^	23.03 ^ab^	11.57 ^abc^	0.04 ^abc^	3.21 ^abc^	33.24 ^a^	18.045 ^b^	0.10	3.65 ^bc^	1.57 ^ab^	0.23 ^ab^	3.23 ^a^
E3	0.22 ^c^	23.03 ^ab^	12.01 ^ab^	0.04 ^bc^	2.70 ^d^	33.03 ^a^	18.10 ^b^	0.12	4.02 ^ab^	1.46 ^b^	0.26 ^a^	3.26 ^a^
SEM	0.008	0.004	0.302	0.004	0.108	0.441	0.322	0.014	0.127	0.070	0.020	0.1246
Main effects												
Diet												
C	0.29 ^a^	23.49	11.76	0.04	2.46 ^c^	31.65 ^b^	20.75 ^a^	0.14 ^b^	4.09 ^a^	0.77 ^b^	0.14 ^b^	2.23 ^b^
E1	0.25 ^b^	23.24	11.37	0.05	2.96 ^b^	33.53 ^a^	17.79 ^b^	0.18 ^a^	3.64 ^b^	1.61 ^a^	0.26 ^a^	3.31 ^a^
E2	0.24 ^b^	23.11	11.36	0.05	3.39 ^a^	33.70 ^a^	17.53 ^b^	0.17 ^ab^	3.56 ^b^	1.70 ^a^	0.22 ^a^	3.20 ^a^
E3	0.25 ^b^	23.10	11.28	0.05	3.11 ^ab^	33.64 ^a^	18.03 ^b^	0.13 ^c^	3.68 ^b^	1.59 ^a^	0.25 ^a^	3.19 ^a^
Time												
2 weeks	0.27 ^a^	23.12	10.94 ^b^	0.05 ^a^	3.23 ^a^	33.82 ^a^	18.28 ^b^	0.16	3.57 ^b^	1.57 ^a^	0.21	2.97
4 weeks	0.24 ^b^	23.35	11.95 ^a^	0.04 ^b^	2.73 ^b^	33.44 ^b^	18.77 ^a^	0.15	3.91 ^a^	1.40 ^b^	0.22	3.00
*p*-Value												
Diet	<0.0001	0.0363	0.3865	0.0967	<0.0001	<0.0001	<0.0001	0.3673	0.0008	<0.0001	<0.0001	<0.0001
Time	<0.0001	0.9116	<0.0001	<0.0001	<0.0001	<0.0001	0.0359	0.0872	0.0006	0.001	0.5921	0.7129
Diet × Time	0.0069	0.9159	0.1603	0.4694	0.028	0.0756	0.3157	0.7799	0.0548	0.4648	0.6387	0.3424

C—control diet; E1—diet supplemented with 6% linseed meal + 2% dried kapia pepper; E2—diet supplemented with 6% linseed meal + 2% dried sea buckthorn pomace; E3—diet supplemented with 6% linseed meal + 2% dried carrot; n = 6; SEM, standard error of the mean; SFA = saturated fatty acids; MUFA = monounsaturated fatty acids; PUFA = polyunsaturated fatty acids^; a–e^ Mean values within a row not sharing the same superscript are significantly different at *p* < 0.05.

**Table 7 foods-10-01246-t007:** Effects of using linseed meal and carotenoid sources on egg yolk unsaturated fatty acids content.

Item	ΣSFA	ΣMUFA	ΣUFA	ΣPUFA	of Which:		Σn6/Σn3	ΣSFA/ΣUFA	ΣPUFA/ΣMUFA
					Σn3	Σn6			
2 weeks									
C	34.91 ^b^	36.42 ^a^	65.09 ^ab^	28.67 ^b^	3.30 ^b^	25.36 ^a^	7.73 ^a^	0.54 ^bc^	0.79 ^b^
E1	34.31 ^b^	37.59 ^a^	65.52 ^ab^	27.94 ^bc^	5.61 ^a^	22.33 ^bc^	3.99 ^b^	0.52 ^bc^	0.75 ^bc^
E2	34.82 ^b^	38.36 ^a^	65.05 ^ab^	26.68 ^c^	5.43 ^a^	22.45 ^bc^	3.92 ^b^	0.53 ^bc^	0.70 ^c^
E3	34.20 ^b^	38.41 ^a^	65.78 ^a^	27.37 ^bc^	5.32 ^a^	22.05 ^bc^	4.03 ^b^	0.52 ^c^	0.72 ^bc^
4 weeks									
C	36.67 ^a^	33.09 ^b^	63.27 ^c^	30.18 ^a^	3.44 ^b^	26.74 ^a^	7.79 ^a^	0.58 ^a^	0.91 ^a^
E1	35.85 ^ab^	36.66 ^a^	64.05 ^bc^	27.39 ^bc^	5.22 ^a^	22.17 ^bc^	4.26 ^b^	0.56 ^ab^	0.75 ^bc^
E2	35.09 ^ab^	37.04 ^a^	64.77 ^abc^	27.73 ^bc^	5.28 ^a^	21.25 ^c^	4.25 ^b^	0.54 ^abc^	0.75 ^bc^
E3	35.50 ^ab^	36.31 ^a^	64.47 ^abc^	28.15 ^bc^	5.26 ^a^	22.90 ^b^	4.37 ^b^	0.55 ^abc^	0.78 ^b^
SEM	0.368	0.502	0.365	0.326	0.118	0.310	0.147	0.009	0.018
Main effects									
Diet									
C	35.79	34.76 ^b^	64.18	29.42 ^a^	3.37 ^b^	26.05 ^a^	7.76 ^a^	0.56	0.85 ^a^
E1	35.08	37.12 ^a^	64.79	27.66 ^b^	5.41 ^a^	22.25 ^b^	4.12 ^b^	0.54	0.75 ^b^
E2	34.96	37.70 ^a^	64.91	27.21 ^b^	5.36 ^a^	21.85 ^b^	4.09 ^b^	0.54	0.72 ^b^
E3	34.85	37.36 ^a^	65.12	27.76 ^b^	5.29 ^a^	22.47 ^b^	4.20 ^b^	0.54	0.75 ^b^
Time									
2 weeks	34.56 ^b^	37.69 ^a^	65.36 ^a^	27.66 ^b^	4.92	22.75 ^b^	4.91 ^b^	0.53 ^b^	0.74 ^b^
4 weeks	35.78 ^a^	35.77 ^b^	64.14 ^b^	28.36 ^a^	4.80	23.56 ^a^	5.17 ^a^	0.56 ^a^	0.80 ^a^
*p*-Value									
Diet	0.0622	<0.0001	0.0773	<0.0001	<0.0001	<0.0001	<0.0001	0.0664	<0.0001
Time	<0.0001	<0.0001	<0.0001	0.0042	0.1717	0.0006	0.0202	<0.0001	<0.0001
Diet × Time	0.2008	0.0992	0.1931	0.0199	0.183	0.0789	0.7489	0.1929	0.0138

C—control diet; E1—diet supplemented with 6% linseed meal + 2% dried kapia pepper; E2—diet supplemented with 6% linseed meal + 2% dried sea buckthorn pomace; E3—diet supplemented with 6% linseed meal + 2% dried carrot; n = 6; ΣSFA = sum of saturated fatty acids; ΣMUFA = sum of monounsaturated fatty acids; PUFA = sum of polyunsaturated fatty acids. Σ n-3 PUFA = C18:3n-3 + C18:4n-3; Σ n-6 PUFA = C18:2n-6 + C20:2n-6 + C20:3n-6 + C20:4n-6; SEM, standard error of the mean. ^a–c^ Mean values within a row not sharing the same superscript are significantly different at *p* < 0.05.

**Table 8 foods-10-01246-t008:** Effects of using linseed meal and carotenoid sources on cholesterol content and lipid oxidative status of the yolk.

Item	Experimental Diets				SEM	*p*-Value
	C	E1	E2	E3		
Fat and cholesterol content of the yolk						
Fat, %	26.21 ^b^	29.16 ^a^	28.38 ^a^	28.00 ^a^	0.323	0.0081
Cholesterol (g/100 g dried yolk)	1.85 ^a^	1.68 ^b^	1.64 ^b^	1.63 ^b^	0.028	0.0081
Lipid oxidative status of the yolk						
PV (meq active oxygen/kg)	0.34 ^b^	0.23 ^a^	0.23 ^a^	0.19 ^a^	0.018	0.0302
CD (µmol/g)	6.40	6.41	6.37	6.31	0.023	0.3962
CT (µmol/g)	2.44 ^bc^	2.46 ^b^	2.38 ^ab^	2.37 ^a^	0.012	0.0181
TBARS (mg MDA/kg)	0.17 ^b^	0.14 ^a^	0.16 ^ab^	0.14 ^a^	0.004	0.0358

C—control diet; E1—diet supplemented with 6% linseed meal + 2% dried kapia pepper; E2—diet supplemented with 6% linseed meal + 2% dried sea buckthorn pomace; E3—diet supplemented with 6% linseed meal + 2% dried carrot; n = 6; PV—peroxide value; CD—conjugated dienes; CT—conjugated trienes; TBARS—thiobarbituric acid reactive substances; SEM, standard error of the mean; ^a,b,c^ Mean values within a row not sharing the same superscripts are significantly different at *p* < 0.05.

## Data Availability

All data is contained within the article.

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
