# Peer review of "Effects of Linseed Meal and Carotenoids from Different Sources on Egg Characteristics, Yolk Fatty Acid and Carotenoid Profile and Lipid Peroxidation"

_foods, 2021, doi:10.3390/foods10061246_

Round 1

Reviewer 1 Report

Review on the Manuscript ID: foods-1201573 entitled „Effects of linseed meal and carotenoids from different sources on egg characteristics, yolk fatty acid and carotenoid profile and lipid peroxidation“

L20 After two respectively four weeks of feeding..- rephrase this.

L25 In conclusion, inclusion of these carotenoid sources- rephrase

L244 Put the name of the column as well as producer and country for capillary column

L 290 Put the producer and country for the HP-5 column

In Table 2 I suggest doing statistical analysis for the proximate composition, fatty acid and carotenoid profile of dried kapia pepper, sea buckthorn pomace and carrot

In table 3 L347 you mention Fischer test results but when describing statistical analysis done you mention that Tuckey test was applied. Correct that

L399 Figure 1 presents the evolution of carotenoids concentration in the fresh yolk samples during the feeding period.- did you do statistical analysis for this? If not, I suggest doing it and put the results of the statistics on the figure 1 and correct accordingly to the results of statistical analysis in the results and discussion section.

L404 Although the lutein content did not change significantly after 3 weeks of feeding- show the statistical results in Figure 1

L551 do not use ppm, rather mg/kg or other SI unit

Author Response

Response to Reviewer 1 Comments

Point 1: L20 After two respectively four weeks of feeding..- rephrase this.

Response 1. We really appreciate the observation. We rephrased the sentence as reviewer suggested (page 1, lines 20-21).

Point 2: L25 In conclusion, inclusion of these carotenoid sources- rephrase

Response 2: We agree the observation made and we rephrased the sentence (page1, lines 26-27).

Point 3: L244 Put the name of the column as well as producer and country for capillary column

Response 3: We included the information required (page 6, lines 246-247).

Point 4: L 290 Put the producer and country for the HP-5 column

Response 4: We included the information required (page 7, line 293-294).

Point 5: In Table 2 I suggest doing statistical analysis for the proximate composition, fatty acid and carotenoid profile of dried kapia pepper, sea buckthorn pomace and carrot

Response 5: We really appreciate the observation. The analyses were performed in triplicate. As you suggested, we inserted in the table the standard deviations for each parameter (page 8, table 2).

Point 6: In table 3 L347 you mention Fischer test results but when describing statistical analysis done you mention that Tuckey test was applied. Correct that

Response 6: We really appreciate the observation. We agree the observation made and we corrected accordingly (page 9, line 350).

Point 7: L399 Figure 1 presents the evolution of carotenoids concentration in the fresh yolk samples during the feeding period.- did you do statistical analysis for this? If not, I suggest doing it and put the results of the statistics on the figure 1 and correct accordingly to the results of statistical analysis in the results and discussion section.

Response 7: I express my sincere thanks for your support and for your constructive suggestions. I have acted upon your recommendations which have resulted in a significant enhancement of the quality of this paper. Thus, we recreated the figure 1, we inserted the statistical significance and we performed the results and discussions accordingly (page 12, results section line 411-420 and discussion section page 20 lines 630-644).

Point 8: L404 Although the lutein content did not change significantly after 3 weeks of feeding- show the statistical results in Figure 1

Response 8: We agree the observation made and we modified accordingly (page 12, results section line 411-420 and discussion section page 20 lines 630-644).

Point 9: L551 do not use ppm, rather mg/kg or other SI unit

Response 9: We modified in the text (page 19, line 571).

Reviewer 2 Report

This manuscript reports a feeding study with laying hens which were fed diets supplemented with linseed meal and various kinds of plant supplements (pepper, sea buckthorn pomace, carrot). The effect of these supplements on laying performance and relevant nutrients, i.e. fatty acids in the eggs were investigated. The study is technical sound. The manuscript is well written and easy to read and understand. There are only few points to be considered:

  1. The concept of supplementing eggs with n-3 PUFA by feeding oil seeds or oils rich in n-3 PUFA is not a new one. Already in the last century, several studies have been published dealing with the effects of feeding linseed meal or linseed oil on laying performance and the fatty acid composition of eggs. Such studies should be considered in the Discussion of the effects on laying performance and fatty acid composition. Examples for these older studies are: Van Elswyk: British Journal of Nutrition 78, 561-569 (1997); Roth-Maier et al.: Agribiological Research 51, 271-275 (1998); Eder et al.: Archiv für Geflügelkunde 62, 223-228 (1998); Jeroch et al. International Symposium on Physiology of Livestock: The Symposium is Devoted to the 10th Anniversary of the Research Center of Digestive Physiology and Pathology Lithuanian Veterinary Academy: 26-27 September 2002.

  2. Table 8: Most of the health lipid indices are not valuable and useful:

AI (atherogenic index): C18:0 is not a pro-atherogenic fatty acid (as it is neutral with respect to plasma cholesterol concentration in humans).

TI (thrombogenic index): saturated fatty acids are not prothrombotic; n-6 PUFA are prothrombotic, n-3 PUFA are anti-thrombotic.

h/H (hypocholesterolemic/hypercholesterolemic): n-3 PUFA are not hypocholesterolemic but are neutral with respect to plasma cholesterol concentration.

Elongase and desaturation indices: These indices can be applied only if there is no difference in the fat source of the diets. The reason for this is that changes in the fatty acid composition could be due to desaturation and elongation processes but also by incorporation of fatty acids from the diet. Enrichment of eggs with n-3 PUFA is surely advantageous from a health point of view. However, the discussion should be limited to beneficial effects of n-3 PUFA.
Overall, I strongly recommend to remove the table and the indices from all parts of the manuscript as these indices do not have any value.

Author Response

Response to Reviewer 2 Comments

Point 1: The concept of supplementing eggs with n-3 PUFA by feeding oil seeds or oils rich in n-3 PUFA is not a new one. Already in the last century, several studies have been published dealing with the effects of feeding linseed meal or linseed oil on laying performance and the fatty acid composition of eggs. Such studies should be considered in the Discussion of the effects on laying performance and fatty acid composition. Examples for these older studies are: Van Elswyk: British Journal of Nutrition 78, 561-569 (1997); Roth-Maier et al.: Agribiological Research 51, 271-275 (1998); Eder et al.: Archiv für Geflügelkunde 62, 223-228 (1998); Jeroch et al. International Symposium on Physiology of Livestock: The Symposium is Devoted to the 10th Anniversary of the Research Center of Digestive Physiology and Pathology Lithuanian Veterinary Academy: 26-27 September 2002.

Response 1:  We really appreciate the observation. We agree the observation made and we included the idea in the manuscript (page 20, line 650-654).

Point 2: Table 8: Most of the health lipid indices are not valuable and useful:

AI (atherogenic index): C18:0 is not a pro-atherogenic fatty acid (as it is neutral with respect to plasma cholesterol concentration in humans).

TI (thrombogenic index): saturated fatty acids are not prothrombotic; n-6 PUFA are prothrombotic, n-3 PUFA are anti-thrombotic.

h/H (hypocholesterolemic/hypercholesterolemic): n-3 PUFA are not hypocholesterolemic but are neutral with respect to plasma cholesterol concentration.

Elongase and desaturation indices: These indices can be applied only if there is no difference in the fat source of the diets. The reason for this is that changes in the fatty acid composition could be due to desaturation and elongation processes but also by incorporation of fatty acids from the diet. Enrichment of eggs with n-3 PUFA is surely advantageous from a health point of view. However, the discussion should be limited to beneficial effects of n-3 PUFA.
Overall, I strongly recommend to remove the table and the indices from all parts of the manuscript as these indices do not have any value.

Response 2: Thank you for the detailed explanation made. We agree the observation and consequently we removed the table 8. Also we deleted in the text as you suggested (material and method section page 6-7 lines 250-255; results section page 17, lines 507-515 and page 18 lines 529-535; discussion section page 21, lines 682-701). Regarding the beneficial effects of n-3 PUFA, we improved the discussion section as you suggested (page 21, lines 674-680).
